# Structural disorder and distinctive motifs in the C-terminal region of the MADS-domain transcription factors are conserved across diverse taxa

Erandi Ramírez-Aguirre[1]◉, Teresa Beatriz Nava-Ramírez[2]◉, Alejandra A. Covarrubias[2]*, Adriana Garay-Arroyo[1]*

**1** Laboratorio de Genética Molecular, Desarrollo y Evolución de Plantas, Depto. de Ecología Funcional, Instituto de Ecología, Universidad Nacional Autónoma de México, Ciudad de México, México,
**2** Departamento de Biología Molecular de Plantas, Instituto de Biotecnología, Universidad Nacional Autónoma de México, Cuernavaca, México

◉ These authors have equal participation in the research and are the first authors.
* agaray@iecologia.unam.mx (AG-A); alejandra.covarrubias@ibt.unam.mx (AAC)

## Abstract

Plants have acquired the ability to adapt and respond to varying environmental conditions through modifications in their developmental programs. This adaptability relies on the plant's capacity to sense environmental cues and respond via diverse signal transduction pathways and transcriptional regulation. Transcription factors are central in these processes, orchestrating specific gene expression in both developmental and stress responses. In *Arabidopsis thaliana*, 91% of transcription factors contain large intrinsically disordered regions (IDRs). The structural flexibility in these regions is critical in protein-protein interactions and contributes to functional versatility across different cell types. MADS-domain transcription factors constitute an eukaryotic protein family involved in a diversity of developmental processes and stress responses. Using bioinformatic tools, we found that most Arabidopsis MADS-domain proteins contain IDRs (≥30 residues) in their C-terminal region, with a higher proportion of global disorder in Type II compared to Type I MADS-domain proteins. Remarkably orthologous proteins from non-plant species in the Eukarya domain (*Drosophila melanogaster*, *Saccharomyces cerevisiae,* and *Homo sapiens*) also present disordered C-terminal regions, containing longer IDRs than those found in Arabidopsis*,* or other analyzed plant species. Additionally, conserved motifs were identified within the C-terminal IDRs of Arabidopsis Type I and Type II MADS-domain proteins, suggesting interactions with co-regulatory partners. We also identified putative activation domains in the C-terminal region of Type I and Type II MADS-domain proteins. The involvement of IDRs in selecting co-regulators is further supported by the identification of Molecular Recognition Features (MoRFs) in Type II MADS-domain proteins. The conserved structural disorder in the C-terminal region of MADS-domain proteins,

**Data availability statement:** All relevant data are within the paper and its Supporting information files.

**Funding:** This work was partially financed by the projects PAPIIT No. IN213524 from Universidad Nacional Autónoma de México (UNAM), and project No. CBF2023- 2024-1002 from the Consejo Nacional de Humanidades, Ciencias y Tecnologías (CONAHCyT) granted to A.G.A.; and projects No. CF-2019/252952 and CF2023-I-503 from the Consejo Nacional de Humanidades, Ciencias y Tecnologías (CONAHCyT) granted to A.A.C.

**Competing interests:** The authors have declared that no competing interests exist.

which includes specific motifs, across diverse domains of life provides valuable insights into their structural properties and mechanisms of action as transcriptional regulators.

## Introduction

Plants have developed a wide range of mechanisms to adapt to changing environmental conditions. Their ability to sense various signals and respond accordingly through diverse and complex signaling pathways to accomplish accurate gene regulation, allows them to exhibit phenotypic plasticity. This plasticity enables plants to optimize their growth and development in response to factors such as light, temperature, water availability, nutrient levels, and biotic stresses. Remarkably different species of plants share plastic traits associated with their specific habitats (e.g., [1]), highlighting the adaptive role of plastic responses and underlying the functional role of the genes involved and their regulation [2].

Transcription factors (TFs) are central regulators in all organisms, playing key roles in diverse biological processes, such as growth, differentiation, hormone signaling, stress responses, and immune defense. By integrating signals from different pathways, TFs orchestrate the complex regulatory networks that underlie plant development and adaptation, enabling plants to respond dynamically to changing environmental conditions [3–9].

MADS-domain TFs belong to a eukaryotic protein family that participates in several developmental processes and stress responses [10,11]. The name MADS-box is derived from the four founding members of this family: *MINICHROMOSOME MAINTENANCE 1* (*MCM1*) in yeast, *AGAMOUS* (*AG*) in *Arabidopsis thaliana* (from now on Arabidopsis), *DEFICIENS* (*DEF*) in *Antirrhinum majus* (snapdragon), and *SERUM RESPONSE FACTOR* (*SRF*) in humans [12]. The MADS (M) domain is highly conserved and present in all members of the family and is typically located in the N-terminal region of the protein. However, there are exceptions to this pattern, such as in SRF in humans and MCM1 in yeast, where the MADS domain is found in different positions within the protein [13,14]. While some organisms have just a few MADS-box genes, such as *Saccharomyces cerevisiae* (yeast) with four, *Drosophila melanogaster* with two, and *Homo sapiens* with five [15–18], plants exhibit a considerably higher abundance of these genes (52–167 in 21 angiosperms) [19–22]. MADS-domain proteins are classified into two main types: Type I (Serum Response Factor (SRF)-like), and Type II (Myocyte Enhancer Factor (MEF)-like) based on their sequence characteristics, genomic organization, and functional roles [23,24]. In yeast, MADS-domain proteins are involved in key processes such as arginine metabolism, osmotic stress response, and mating type regulation. In *Drosophila*, MEF2 genes participate in muscle differentiation, while in humans the different MEF2 genes also contribute to heart and neural development, and the response to various diseases [13,16,25–27]. In plants, MADS-domain proteins play several roles in development, encompassing the transition to flowering, flower organ

development, fruit ripening, root development, and vegetative phase change, among others [24]. In Arabidopsis, Type I MADS-domain proteins have been primarily implicated in female gametogenesis and seed development, while Type II MADS-domain proteins play central roles in controlling floral organ identity and flower development in angiosperms and are involved in almost all Arabidopsis developmental processes [24,28]. Type I MADS-domain proteins typically consist of three domains: M, I (intervening), and C (C-terminal domain), whereas Type II proteins possess four domains, forming the acronym MIKC [29–33]. In both cases, the "M" domain is responsible for DNA binding but also participates in protein-protein interactions. MADS-domain proteins bind to a specific DNA sequence known as the "CArG" box located in the regulatory regions of target genes. The "I" domain is located between the "M" and the "K" domains. While it is less conserved than the "M" domain, it also participates in DNA-binding and dimerization specificity [33,34]. The "K" domain derives its name from its structural resemblance to proteins known as keratins. This domain is involved in protein-protein interactions and contributes to the formation of higher-order complexes. Finally, the "C" domain is located at the C-terminal of the protein and exhibits variations in length and sequence among different MADS-domain proteins. For some MADS-domain proteins, it has been shown that this domain participates in transactivation and in the formation of higher-order complexes [35].

A phylogenetic analysis of 107 Arabidopsis MADS-box sequences revealed that these genes can be grouped into two main lineages (Type I and Type II) or five subfamilies: Mα, Mβ, and Mγ, and MIKC$^c$ and MIKC*. The Mα, Mβ, and Mγ, subfamilies belong to Type I MADS-domain proteins whereas the MIKCc and MIKC* subfamilies are classified as Type II [24,36]. The genomic distribution of MIKC genes, along with evidence from genome history, suggests that these genes existed before the Arabidopsis genome polyploidization event and are distinct from the Mα, Mβ, and Mγ subfamilies [23,36].

In Arabidopsis, 91% of TFs are characterized by large intrinsically disordered regions (IDRs), crucial for diverse cellular functions [37]. IDRs are segments of proteins that lack a fixed or stable three-dimensional structure under physiological conditions. Their unique physicochemical properties arise from the specific nature of their amino acid sequences and the characteristics of the individual amino acid residues within these sequences. Intrinsically disordered proteins (IDPs) and/or IDRs often lack a significant number of hydrophobic residues typically associated with folded protein domains whereas they are enriched in amino acids that allow their flexibility [38]. The IDP amino acid sequences enable a structural dynamic that depends on the physicochemical properties of their environment and the interactions with other molecules [39,40]. This flexibility plays a pivotal role in mediating IDP multiple protein-protein interactions and in their participation in different developmental processes, such as cell cycle, transcriptional control, and responses to different stress conditions [41,42]. However, only a limited number of TF families have been analyzed, highlighting the presence of IDRs and their association with their role in their respective functions [43–47].

Interestingly, IDRs often contain specific sequences known as Molecular Recognition Features (MoRFs), these short transiently folding sequences play a key role in determining site-partner specificity [48,49]. MoRFs show distinctive physicochemical characteristics that may aid in protein interaction through the hydrophobicity of their amino acid composition (rich in proline and methionine) and the hydrophilic nature of IDRs [50,51]. The charge of individual amino acids and their electrostatic interactions affect the conformational structure of the protein, which in turn affects its binding specificity and stability. This has led to the hypothesis that MoRFs are key participants in the multi-partner binding capacity of hub proteins [52–54]. The dynamic interconnectivity of IDPs/IDRs has also been associated with their ability to aggregate through liquid-liquid phase separation (LLPS), a process that plays a critical role in protein and RNA organization [55]. LLPS is now recognized as a fundamental organizational and regulatory principle across all organisms. It enables the concentration of specific proteins and nucleic acids into biomolecular condensates, i.e., membrane-less organelles that regulate cellular activity by localizing particular proteins, accelerating enzymatic reactions, and favoring selective interactions. This dynamic regulation supports the precise control of a diversity of processes involved in growth, development, and responses to environmental changes and pathogens [56].

In this work, we used bioinformatic tools and publicly available datasets to investigate the presence of IDRs in MADS-domain proteins from plants and other taxa. We found that Arabidopsis Type I and Type II MADS-domain TFs present a high level of disorder, ranging from 20 to 80%, with longer IDRs in their C-terminal domain. This characteristic was found to be conserved in orthologs of these proteins from other plant species as well as from *Drosophila melanogaster*, *Saccharomyces cerevisiae,* and *Homo sapiens*. Of note, within the C-terminal region of proteins in both the SOC1 and the FLC clades [23,24], we found two different motifs common to all members of the SOC1 and FLC clades, suggesting functional restriction and phylogenetic conservation within these groups. Interestingly, the SOC1 motif is conserved across SOC1 orthologs from different plant species, and in other Arabidopsis Type II MADS-domain proteins. Many Arabidopsis MADS-domain proteins also contain MoRFs that overlap with or are located just a few amino acids away from the SOC1 and the FLC motifs in the last residues of the C-terminal region. We mapped putative activation domains (ADs) across all clades of Type I and Type II MADS-domain proteins on their C-terminal region. Interestingly, Type I showed a higher proportion of ADs compared to Type II MADS-domain proteins. Finally, our *in silico* analysis predicts that some MADS-domain proteins may form condensates, potentially leading to different conformational arrangements and expanding their functional roles. The information in this work adds valuable insights to better understand the MADS TFs molecular mechanisms involved in the control of plant growth, morphogenesis, and responses to environmental changes.

## Materials and methods

### Complete sequence retrieval and domain description

Individual protein sequences of *Arabidopsis thaliana* were retrieved from the TAIR database [57] except for STK (AGL11) which was retrieved from UniProt (Q38836). Protein domains were identified using the UniProt database [58]. The I domain was defined based on the *AtSOC1* protein obtained from the supplementary information in Lai et al. (2021) [33] for both Type I and Type II MADS-domain proteins. In this study, the C-terminal region was defined as the sequence located downstream of the I (for both Type I MADS-domain proteins) or downstream of the K domain (for Type II MADS-domain proteins). Retrieval of protein sequences from other plants and non-plant organisms was obtained from GenBank, the Rice Annotation Project DataBase, and UniProt [58–60] (S1 Table). A total of 143 sequences were collected, including 100 from Arabidopsis, nine from rice, and at least one homolog from basal plants and other angiosperms (S1 Table). The selection of the Type I MADS-domain proteins was based on Bemer et al. (2010) [28], while the selection of rice sequences covering both Types of MADS-domain protein orthologs was based on Arora et al. (2007) [61].

### Sequence alignment and phylogenetic analysis

After retrieving the individual protein sequences, they were aligned with the MAFFT algorithm at the MAFFT web server [62,63]. The parameters used were L-INS-i and the mafft-homolog function with UniRef. After the alignment of the 143 sequences, we visually inspected for ambiguous misalignments with the AliView program [64]. Using this alignment, we determined the position of the I domain for Type I and Type II MADS-domain proteins. The M and the K domains for all MADS-domain proteins were retrieved from Uniprot. This approach allowed us to accurately map the C-terminal region along with its corresponding IDRs. By assigning approximately 190 amino acid residues up to the K-domain, we found a better alignment of conserved regions among the selected protein sequences. The alignment with the MIK domains was used to recover a phylogeny by applying a Maximum Likelihood approach with the RAxML algorithm, following a JTT substitution model, at the CIPRES website [65–68].

### Motif and Molecular Recognition Features (MoRFs) identification

The MEME algorithm [69,70] was used to identify motifs within the whole protein sequence or the C-terminal regions. Molecular Recognition Features (MoRFs) are small IDRs potentially involved in the initial events of molecular recognition

during protein-protein interactions. These regions undergo disorder-to-order transitions upon binding. MoRFs were identified in the complete protein sequences using the fMoRFpred tool that uses the physicochemical properties of amino acid residues to fit a Support Vector Machine (SVM) model for predicting the presence of MoRFs within IDRs [49,71]. To map the predicted MoRFs, their locations were aligned across the protein sequences alongside the MEME motifs.

## Mapping of activation domains within the C-terminal region of MADS-domain proteins

Potential activation domains (ADs) in MADS-domain proteins were retrieved directly from Morffy et al. (2024) [72] (S2 Table). In this study, the authors experimentally identified activation domains in various plant TFs using a comprehensive library. This library consisted of overlapping 40 amino acid fragments, spanning the entire set of plant TFs, with a step size of 10 amino acids, resulting in a total of 68,441 fragments. These fragments were screened in yeast to assess their transcriptional activation capacity. Based on this experiment and subsequent normalization, a (Plant Activation Domain Identification) PADI score was assigned to each fragment. Among the fragments showing transcriptional activation activity, some corresponded to MADS-domain TFs. The PADI scores included in our manuscript are those reported here, and the localization of activation domains (AD) within MADS-domain TFs was inferred from the results obtained in Morffy et al. (2024) [72], where the authors applied a neural network-based algorithm, known as transcriptional activation domain activity (TADA) network. Their work integrated multiple layers of analysis, including the construction of a feature matrix and the use of methods to assess the impact of both individual input features and border local and global interactions predicted by TADA. Additionally, these results were further analyzed using deep learning to identify key properties relevant to the prediction of ADs. These authors also applied a Shapley additive explanations (SHAP) analysis to capture non-linear and linear correlations, thereby uncovering complex patterns. This was followed by additional deep-learning steps that culminated in the development of a tool capable of predicting potential ADs. Using the Morffy et al. (2024) [72] dataset, we filtered the information for MADS-domain proteins to localize the "NOT-AD" and the "AD" fragments within their C-terminal IDR.

## IDR prediction and structural disorder score of the complete proteins

The majority of Arabidopsis IDRs were retrieved from the Alphafold section of the MobiDb database [73] (accessed in November 2023). The IDR prediction was conducted with the AlphaFold Colab Notebook with default values or directly in the AlphaFold web server [74] (accessed in January 2024). The resulting structures were visually inspected with the *pbd. file* coupled with the Predicted Aligned Error (PAE) of the proteins in the Chimera X molecular visualizer [75]. We also checked the IDRs for each prediction in JalView and accounted for the low Temperature Factor regions, which correspond to the low pLDDT values [76]. Disordered regions predicted by AlphaFold are those with a pLDDT < 50, usually seen as ribbon-like structures [77, 78]. The structural disorder analysis was assessed using the RIDAO platform [79], which includes various intrinsic disorder predictors [79,80]. Protein sequences in FASTA format were used as input for the analysis, the amino acid sequence of the C-terminal region was manually extracted from the original full-length protein sequences and formatted in FASTA. The RIDAO platform provides two key metrics for each protein: The Average Disorder Score (ADS) and the Percent of Predicted Intrinsic Disorder Residues (PPIDR). The ADS indicate the overall propensity of a protein to be intrinsically disordered, allowing a comprehensive analysis of structural disorder. The PPIDR represents the proportion of amino acids predicted to be disordered, considering those with significant intrinsic disorder score (> 0.5), relative to the total number of residues in the protein [81].

## Physicochemical properties of disordered proteins

Parameters related to the amino acid residue charge (NCPR: net charge per residue, and FCR: fraction of charge residues) and their distribution (patterning parameter kappa) [52] across the C-terminal regions of MADS-domain proteins were calculated using tools available in the CIDER web server [82,83].

### Post-translational modifications (phosphorylation)

Phosphorylation in serine, threonine, and tyrosine residues within the complete protein sequences was predicted using the NetPhos algorithm [84–86]. Additionally, experimentally validated data were obtained from the ATHENA at [87] (accessed in February 2024) and the EPSD databases (accessed in May 2025) [88].

### Condensate formation tendency

Condensate propensity was calculated for the complete MADS-domain protein and their C-terminal domain region sequences using the FuzDrop algorithm [89,90]. Protein sequences in FASTA format were used as input for the analysis, the amino acid sequence of the C-terminal region was manually extracted from the original full-length protein sequences and formatted in FASTA.

### Data analyses and image rendering

We evaluated differences in disorder ADS and PPIDRS among MADS-domain proteins, with a Wilcoxon test under a permutation approach with the coin R package [91]. Also, we compared the raw number of phosphorylation sites among Arabidopsis MADS-domain protein types with a Chi-squared test. All the analyses and graphics were performed in the R program [92]. Protein diagrams were made with the drawProteins library from the Bioconductor repository, and the final rendering was done with the GIMP program [93] (license GPLv3. Version 3.0.4, Free Software). Statistical graphs were made with the tidyverse libraries and with ggplot2 side-by-side working libraries (S3 Table). Other R packages used here are cited in S3 Table.

## Results

### The C-terminal domain of Arabidopsis MADS-domain transcription factor family presents structural disorder propensity

The function of MADS-domain proteins in development is conserved across diverse plant species and other taxa. Given the diversity of cell types and conditions in which these transcription factors regulate gene expression, the structural disorder appears to be an advantageous property, enabling efficient and versatile functionality. In this study, we evaluated the occurrence of structural disorder in 58 Type I and 42 Type II MADS-domain proteins from Arabidopsis to gain insight into their protein structure and its relationship to their function. To evaluate the degree of disorder, the Average Disorder Score (ADS) and the Percent of Predicted Intrinsic Disorder Residues (PPIDR) were calculated in the RIDAO platform, for both the complete proteins and their C-terminal regions. For Type I MADS-domain proteins, both the mean ADS and mean PPIDR of the complete proteins and their C-terminal regions are similar (Fig 1A and 1C, left panel). In contrast, for Type II MADS-domain proteins, both the mean ADS and mean PPIDR are larger for the C-terminal region than for the complete proteins (Fig 1A and 1C, right panel). Also, the relation between the protein length and ADS or PPIDR values revealed a broader length distribution (100–450 amino acid residues) for Type I MADS-domain proteins and only a minor inverse correlation between protein length and global disorder (Fig 1B and 1D, left panel). In contrast, Type II proteins exhibited a narrower length range (200–300 amino acid residues) but followed a positive trend between length and disorder (Fig 1B and 1D, right panel). Moreover, Type II MADS-domain proteins present a significantly higher disorder (ADS and PPIDR) compared to Type I proteins (permutation Wilcoxon test, [ADS] $Z = -3.1497$, $p = 0.0016$; [PPIDR] $Z = -3.897$ $p = 1\times10^{-4}$, Fig 1). AlphaFold-based analysis of the structural disorder distribution revealed that this property is predominantly located in the C-terminal regions for both types: in Type I this region follows the I-domain, while in Type II proteins it is found beyond the K-domain (Fig 2A, S4 Table). Notably, the longest IDRs in both types were consistently located within the C-terminal region, and the IDRs at the C-terminal region in Type I MADS-domain proteins were longer than in the Type II MADS-domain proteins (permutation Wilcoxon test, $Z = 4.1552$, p-value $< 1e-04$p, Fig 2, S4 Table). The number of IDRs containing

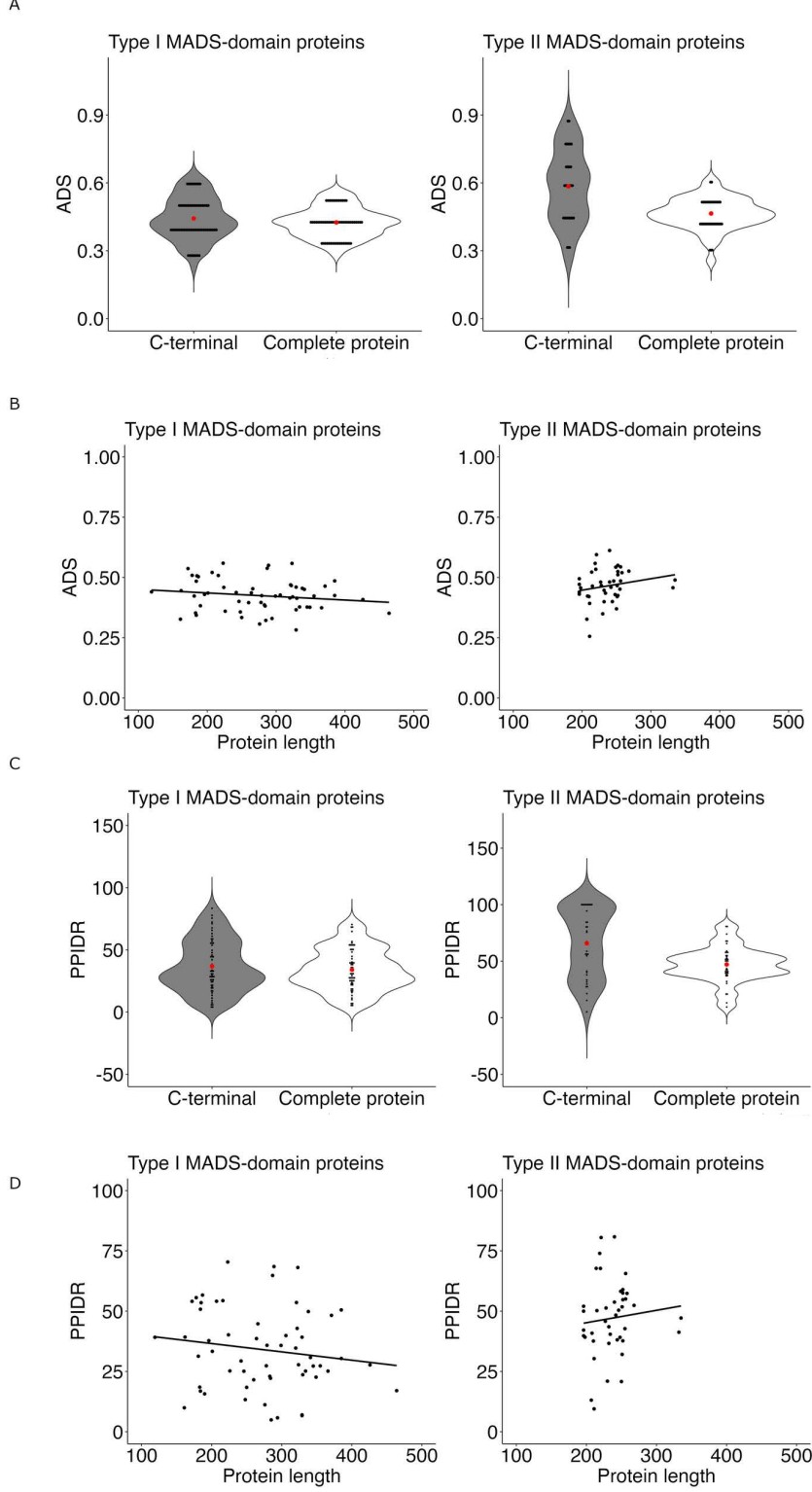

**Fig 1. The Average Disorder Score (ADS) and Predicted Percentage of Intrinsically Disordered Residues (PPIDR) values in Type I and Type II MADS-domain proteins. (A)** Distribution of ADS in MADS-domain TFs. **(B)** Scatterplots showing the association between protein length and ADS. **(C)** Distribution of PPIDR values in MADS-domain TFs. **(D)** Scatterplots showing the association between protein length and PPIDR. Red dots indicate

the mean of each value, and black dots indicate the raw values of either ADS or PPIDR per protein. Statistical analysis using the Wilcoxon test shows a significant difference between Type I and Type II proteins (p = 0.0002).

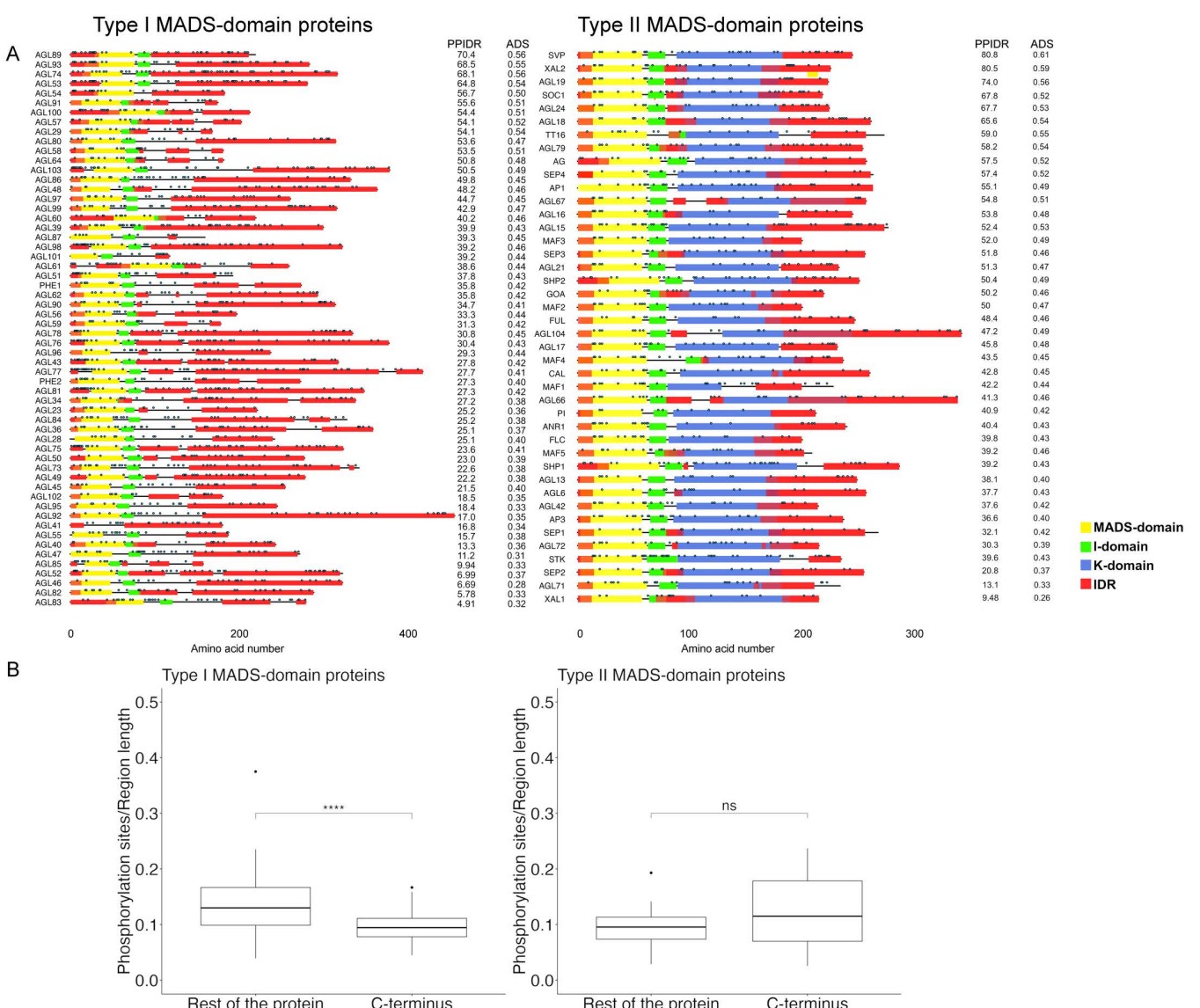

**Fig 2. MIKC domains, disordered regions, and phosphorylation sites for MADS-domain proteins. (A)** Schematic representation of Type I and Type II MADS-domain proteins indicating their distinctive domains: MADS (M = yellow), Intervening (I = green), Keratin-like (K = blue), and Intrinsically Disordered Regions (IDR = red). Numbers below indicate the amino acid positions. Dots above the domains indicate putative phosphorylation sites. The columns to the right of each protein diagram indicate their corresponding values for PPIDR and ADS. Proteins are arranged in descending order according to their PPIDR. **(B)** Distribution of predicted phosphorylation sites in the N-terminal and C-terminal regions of Type I and Type II MADS-domain proteins. Whiskers indicate ±1.5 times the interquartile range (IQR) according to the Tukey method; the middle line denotes the median.

at least 30 amino acid residues was similar between the two groups (S4 Table, Type I total count = 51, Type II total count = 42, $\chi^2$ = 0.87, df = 1, p = 0.351). Nevertheless, in nine Type I MADS-domain proteins (AGAMOUS-LIKE91 [AGL91], AGL29, AGL58, AGL64, AGL87, AGL101, AGL59, AGL102, AGL85) we only detected IDRs shorter than 30 amino acid residues, whereas all Type II MADS-domain proteins contain at least one IDR with ≥30 amino acids. Interestingly, most MADS-domain proteins present an IDR at the beginning of the N-terminal, in the first residues of the M-domain.

The flexibility and dimensions of IDRs depend on their amino acid composition, where the charge content and hydrophobicity are key factors. Because the composition of most IDPs includes positive and negative charges, some of their characteristics can be described by the fraction of charged residues (FCR) and net charge per residue (NCPR). Nevertheless, the best descriptors for IDP conformational properties are the FCR and the distribution of oppositely charged residues, defined by a patterning parameter named kappa (k) [52]. On average, polypeptides with low kappa-values (closer to 0) are predicted to adopt more extended conformations, while sequences with higher kappa-values (closer to 1) are expected to form more compact, hairpin-like structures [52]. The kappa-values obtained for the C-terminal of MADS-domain proteins showed that these regions in Type I and Type II proteins are prone to adopting extended conformations, likely due to the counterbalance of the interchain electrostatic interactions resulting from the more even distribution of oppositely charged residues. This result aligns with a mean NCPR close to zero and a mean FCR value at the boundary between weak and strong polyampholytes obtained for these IDRs (S1 Fig). Although this conformational analysis was applied to the IDRs in isolation from the rest of the protein and under certain conditions, which may influence their overall conformational properties, these correlations further reinforce the potential impact of these physicochemical properties on their structural organization and dimensions.

## The C-terminal region of Arabidopsis MADS-domain proteins has a high propensity for phosphorylation

As phosphorylation is closely associated with protein disorder [94], we predicted the phosphorylation propensity of Type I and Type II MADS-domain proteins using the NetPhos algorithm. When comparing the C-terminal sequences of both MADS-domain protein types, we found that the Type II C-terminal contains a similar number of phosphorylation sites compared to those of the Type I C-terminal region (number of phosphorylation sites/ C-terminal length, permutation Wilcoxon test, Z = −1.8163, p = 0.069) (Fig 2B). In contrast, Type I MADS-domain proteins exhibited more abundance of predicted phosphorylation sites in regions outside of their C-terminal region (Fig 2B).

To further investigate the significance of the phosphorylation sites between the N-terminal and the C-terminal regions of MADS-domain proteins, we examined experimentally validated phosphorylation sites in Arabidopsis using two actively curated databases, ATHENA and EPSD. Compared with the large number of predicted sites from NetPhos, experimentally confirmed sites are less represented in Arabidopsis MADS-domain proteins. For Type I MADS-domain proteins, we found 48 experimentally verified sites in the N-terminal region and 17 sites in the C-terminal region. For Type II MADS-domain proteins, there were 18 sites in the N-terminal and 13 in the C-terminal regions (S5 Table).

## The disordered C-terminal is a conserved feature of MADS-domain proteins across diverse taxa

Type I and Type II Arabidopsis MADS-domain proteins contain two or three domains, respectively, which we grouped into the N-terminal region (Fig 2A). As both types have a large, disordered C-terminal region (Fig 2A), we investigated whether this characteristic is more broadly conserved across organisms within the Eukarya domain. As a proof of concept, we only examined MADS-domain proteins from three of the four kingdoms within the Eukarya domain: Plantae, Fungi, and Animalia [95]. We selected sequences from *Homo sapiens* [5]*, Drosophila melanogaster* [2], and *Saccharomyces cerevisiae* [4] and used Arabidopsis MADS-domain protein sequences as a reference due to their extensive characterization. These sequences were compared with those of various phyla within Plantae including Chlorophytes, Charophytes, Gymnospermae, and Angiospermae. We extended this comparison to model species within Animalia (Chordata and Arthropoda) and Fungi (Ascomycota) (Fig 3 and S2 Fig). This analysis showed that all the selected MADS-domain proteins of non-plant

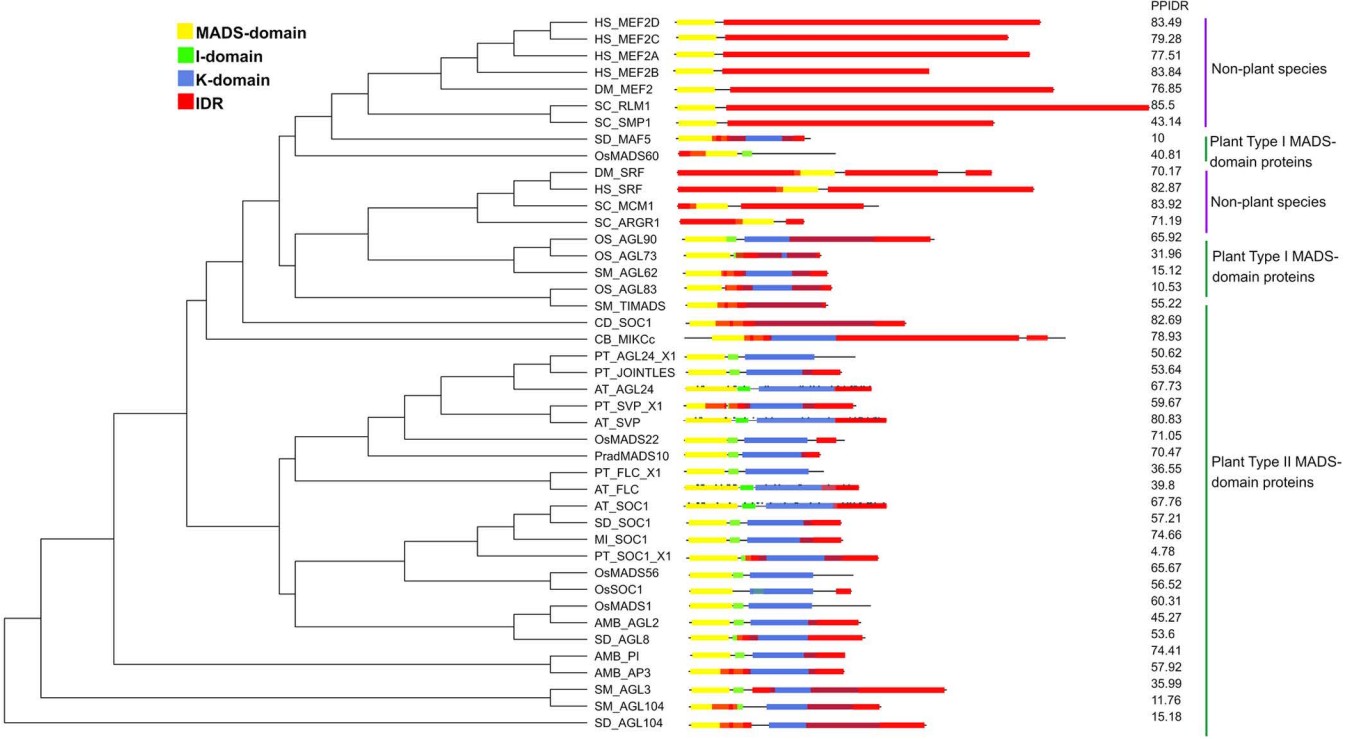

**Fig 3. Intrinsically disordered regions (IDRs) in selected MADS-domain proteins from plant and non-plant species.** The phylogenetic tree presented here is a simplified and rescaled version of the original presented in Supplementary Fig 2, with branch lengths adjusted for clarity. Deeper nodes within Type I MADS-domain protein clades exhibit low phylogenetic support, and their relationships should be interpreted with caution. In contrast, the more derived nodes, particularly among Type II proteins, show stronger phylogenetic support, consistent with previously published phylogenies. PPIDR and ADS values for each protein are shown in the columns to the right of the diagram.

species contain a disordered C-terminal region and that at least for these proteins, these regions are considerably longer than in most plant MADS-domain proteins (Fig 3 and S2 Fig). Moreover, regardless of the taxonomic group to which a species belongs, the C-terminal region consistently presents a higher level of disorder compared to the full-length protein, either with ADS or PPIDR (S3 Fig).

## The disordered C-terminal regions of Arabidopsis MADS-domain proteins contain conserved motifs and MoRFs

To further characterize the C-terminal region of the MADS-domain proteins, we looked for conserved motifs within this region and identified three distinct motifs both in Type I and in Type II MADS-domain proteins (Fig 4). Among the Arabidopsis MADS-domain proteins, there is a conserved pattern of motif distribution, with motifs shared in a subfamily-specific manner. Within the β subfamily of Type I MADS-domain proteins [24], 10 out of 19 proteins (Fig 4) contain both motif 1 and motif 2, whereas seven proteins from both the γ and β subfamilies possess only motif 2 (Fig 4). In the γ subfamily [24], 10 out of 17 proteins have motif 3 (Fig 4). Our analysis shows that none of the α subfamily of Type I MADS-domain proteins contain any distinctive motif.

Among Type II MADS-domain proteins, three motifs are shared by members of different clades. In the SOC1 clade (SOC1, XAL2, AGL42, AGL19, AGL71 and AGL72), all members contain motif 4, which is not only shared by the SOC1 clade members but also by other proteins outside this clade, including SEP3, AGL24, SEP2, AGL15, and SVP (Fig 4, right panel). Within the FLC clade, all members shared motif 5, suggesting functional constraints and convergent acquisition

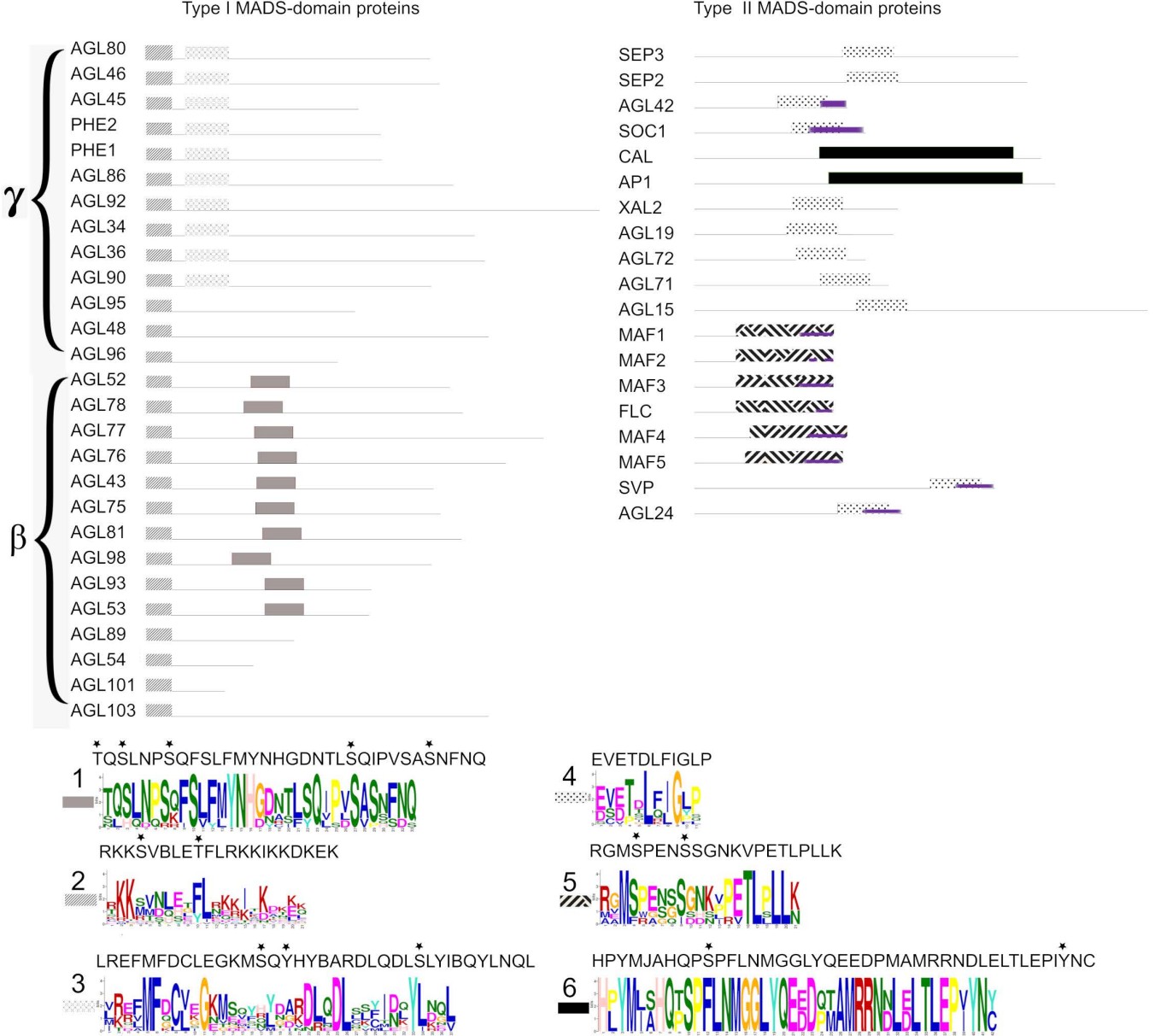

**Fig 4. Motifs identified within the IDRs of the C-terminal regions of MADS-domain proteins.** Motifs were detected using the MEME algorithm [121] based on the C-terminal sequences of Type I and (left panel) Type II MADS-domain proteins (right panel). Type I MADS-domain proteins do not contain molecular recognition features (MoRFs; purple rectangles) near or within the identified motifs. In contrast, in Type II MADS-domain proteins, MoRFs (purple rectangles) overlap with the predicted motifs, supporting their potential functional relevance. MorRFs were predicted using the fMoRF algorithm [123]. Consensus sequences of the identified motifs shown in panels (lower panel).

(Fig 4). Motif 6 is exclusive to CAL and AP1, which are partially redundant in floral meristem determination and belong to the AP1 clade [96]. Additionally, we selected various SOC1 orthologs and members of the FLC clade to further characterize their conserved motifs. For SOC1 orthologs, the consensus motif is predicted to span 19 residues, with a core of 10 highly conserved residues across angiosperms, except in *Populus trichocarpa* (S4A Fig). The FLC consensus motif is 21 amino acids long and contains at least eight conserved residues , primarily located in the latter portion of the sequence [24] (S4B Fig).

We also evaluated the Molecular Recognition Features (MoRFs) across the full-length MADS-domain proteins to determine whether the ubiquitous IDRs in the C-terminal coincide with any MoRFs. Since MoRFs are known to mediate molecular recognition and are proposed to facilitate specific interactions among proteins, we hypothesized that the C-terminal would contain at least one predicted MoRF. Indeed, we found that several MADS-domain proteins, regardless of their species of origin, have a MoRF within the last 10 amino acids of their C-terminal region (S5 Fig). Moreover, several MADS-domain proteins show 1–4 amino acid MoRFs at the beginning of their N-terminal region. Intriguingly, one of these proteins is AG, which also has an IDR of at least 30 amino acids in the N-terminal region. Interestingly, the MoRFs in SOC1, AGL24, AGL42 and SVP proteins were found associated with the SOC1 motif (Fig 4). Similarly, the FLC motif also shows predicted MoRFs within the last 5–7 amino acids, located towards the end of the motif in all the proteins of the FLC clade. In contrast, Type I MADS-domain proteins do not present any MoRFs coinciding with identified motifs (Fig 4).

## The disordered C-terminal regions of MADS-domain proteins contain potential activation domains

Activation domains (AD) in TFs play a central role in the function of these proteins, as they constitute the recruiting sites of coactivator complexes to activate transcription [97]. A connection between IDRs and ADs has been established in several transcription factor families, highlighting the importance of structural flexibility in recognizing diverse molecular targets according to the cellular conditions [98,99]. The highly conserved presence of a C-terminal IDR in Type I and Type II MADS-domain proteins prompted us to search for potential ADs in this region. Using the plant activation domain identification (PADI) score developed by Morffy et al. (2024) [72] to analyze these C-terminal regions, we identified high-scoring regions in 34 (58.3%) of Type I MADS-domain proteins, indicating the presence of a high proportion of potential ADs, compared to the 9 (22.5%) of Type II MADS-domain proteins with a high PADI score (Fig 5 and S2 Table).

Although not all MADS-domain proteins exhibited potential ADs, numerous regions with a high PADI score were found in some of them. To look for possible AD distribution patterns, we graphed the localization of the high PADI scoring regions across the C-terminal region of the MADS-domain proteins. Given the large amount of Type I proteins, we grouped them by subfamilies (Mα, Mβ, and Mγ) as described for Type I MADS-domain proteins to improve clarity in the analysis [24,36]. The C-terminal region of Type I Mα MADS-domain proteins showed a higher abundance of potential ADs (208–341). In contrast, the highest abundance of putative ADs in Type I Mβ was found from 111 to 324 amino acid residues. For the Type I Mγ subfamily, the highest AD abundance was found in two distinct sections (161–278 and 281–339) of their central region. In the case of Type II MADS-domain proteins, potential ADs were more evenly distributed across the region between 171–255 amino acid residues (Fig 5 and S2 Table).

Additionally, for Type I MADS-domain proteins, we found that putative ADs are associated with motifs previously identified in one member of the Mβ subfamily (AGL81), and one member of the Mγ subfamily (PHE1) [36]. Similarly, for Type II MADS-domain proteins, some ADs are associated with different motifs identified in one member of the SOC clade (AGL19), in two members of the SQUA clade (CAL, AP1), and two members of the SEP clade (SEP2 and SEP3) [24]. The overlap of specific motifs within particular phylogenetic groups of MADS-domain proteins supports the functional significance of these potential ADs.

## MADS-domain proteins show a propensity for liquid-liquid phase separation

Some proteins can form compartments within cells where certain proteins, RNA, and metabolites concentrate to orchestrate specific cellular processes. These compartments are generated via Liquid-Liquid Phase Separation (LLPS), a process in which certain molecules concentrate to form a new liquid phase distinct from the surroundings. Some proteins are considered droplet drivers according to their likelihood of forming a droplet state via pLLPS. In this state, protein interactions can occur in different binding configurations, making IDPs common components of these structures. To explore the propensity of MADS-domain proteins to spontaneously undergo LLPS and form condensed cellular states, we used the

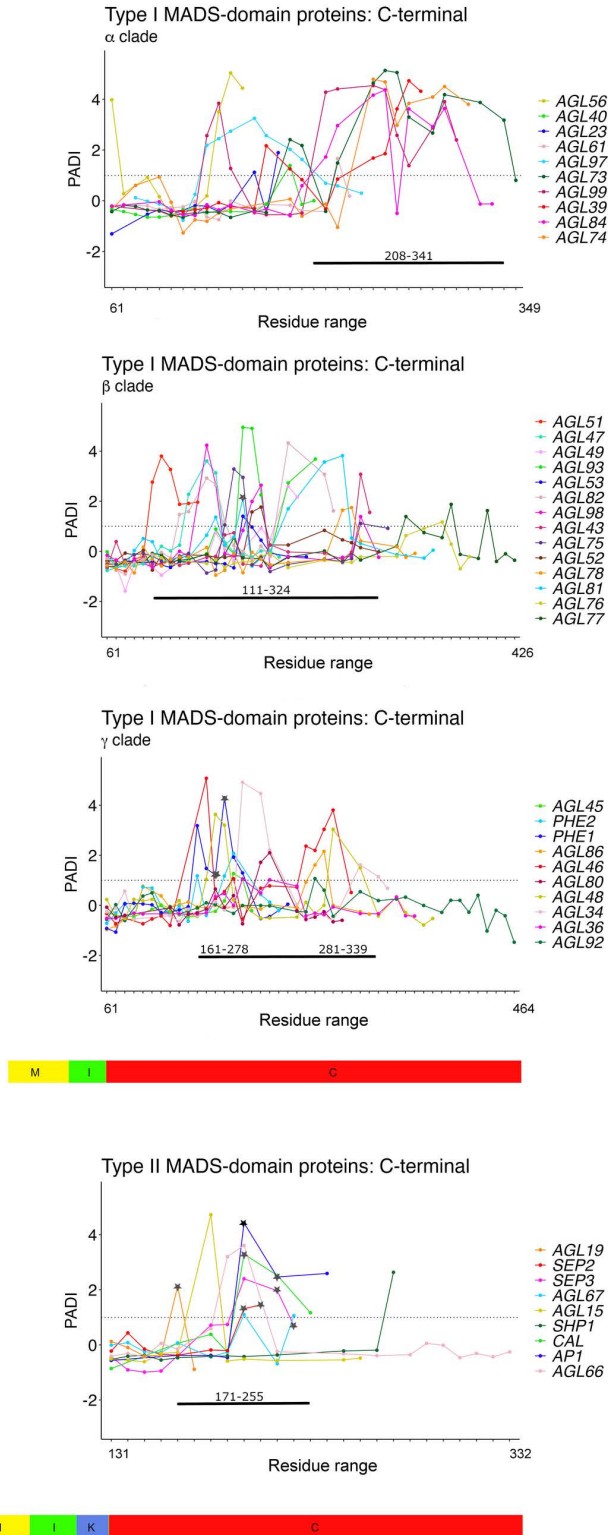

**Fig 5. Potential activation domains in the C-terminal region of MADS-domain proteins.** The Predictive Activation Domain Index (PADI), or scaled activation score, predicts the likelihood of putative activation domains (ADs), with fragments with a PADI score ≥1 classified as potential ADs [64]. In the graphs, dots indicate the positions of 40-amino-acid fragments predicted to containADs, while black stars mark experimentally validated ADs. Numbers

along the horizontal lines show regions within the protein that are enriched in putative ADs. Colored boxes represent the defined domains in Type I and Type II MADS-domain proteins: MADS domain (yellow), I domain (green), K domain (blue), and C-terminal region (red).

FuzDrop server [89]. The FuzDrop algorithm defines a probability threshold value ($p_{LLPS} \geq 0.60$) to identify proteins capable of phase separation and driving droplet formation. This analysis showed that among Type I MADS-domain proteins and their C-terminal domain, the LLPS propensity is widely distributed between proteins with low or high ADS or PPIDR, finding 21 proteins with LLPS propensity (AGL103, AGL93, AGL89, AGL53, AGL74, AGL48, AGL77, AGL102, AGL60, AGL64, AGL23, AGL56, AGL92, AGL98, AGL75, AGL52, AGL76, AGL45, AGL81, AGL43, AGL29), and three more when analyzed only the C-terminal region (AGL91, AGL99, AGL86), except six of the first proteins (AGL60, AGL64, AGL98, AGL52, AGL76, AGL81) (Fig 6 and S4 Table). Regarding Type II MADS-domain proteins, only four proteins showed LLPS propensity (AGL104, AGL79, AGL66, GOA), whose ADS are between 0.4 and 0.6 (S4 Table). This number increased to 17 more when only the C-terminal region was analyzed (including SEP1, SEP4, AGL13, AGL15, AGL18, FCL, MAF1, AP1, AG, AGL19, AGL67, AGL24, SHP2, MAF5, AGL17, AGL72, MAF4) (Fig 6 and S4 Table).

## Discussion

The Arabidopsis MADS-domain proteins participate in nearly all developmental processes and are also involved in many different stress responses [11,29,100]. These proteins are divided into two groups based on their phylogenetic relationship and protein domains: Type I with three domains (M, I, and C), and Type II with four domains (M, I, K, and C) [24,35,101,102].

In this work, we demonstrated through *in silico* analyses that the C-terminal regions of 100 Arabidopsis MADS-domain proteins listed in UniProt are enriched with IDRs (≥30 residues), with no significant differences in the number of IDRs between Type I and Type II proteins. Although we haven't been able to find information regarding the functional characterization of these regions in Type I proteins, several examples underscore the significance of the C-terminal domain in the function of Type II MADS-domain proteins. In particular, specific phenotypes and altered protein-protein interactions in Type II MADS-domain proteins have been associated with point mutations or deletions within IDRs in the C-terminal region, highlighting their functional importance (Fig 7 and S6 Table). For instance, in Arabidopsis, *Raphanus sativus, Nicotiana sylvestris* and *N. tabacum*, the C-terminal regions of *APETALA 1* (*AP1*) and its orthologs have been shown to mediate transcriptional activation in yeast and mammalian cells [101]. Additionally, three *AP1* loss-of-function mutants (*ap1–4, ap1–6* and *ap1–8*), all with mutations in the C-terminal region, exhibit different phenotypes [103]. Similarly, the C-terminal domains of GLOBOSA (GLO) and DEFICIENS (DEF) in *Antirrhinum majus* are critical for the interaction between GLO and DEF and between DEF and SQUAMOSA (SQUA) [103]. Furthermore, the C-terminal domain plays a central role in mediating interactions between MADS-domain proteins and non-MADS-domain proteins. For instance, the co-repressors SEUSS (SEU) and LEUNIG (LUG) interact with AP1 or SEP3 through their C-terminal domains [104]. In Arabidopsis, the K and C-terminal domains of AGAMOUS (AG) are also indispensable for DNA binding [105,106]. Interestingly, a small IDR in the N-terminal of AG, located before the MADS-domain (Fig 1), is essential for its function, as constructs lacking this IDR exhibit a phenotype like an *AP2* mutant. Moreover, overexpression of AG without its C-terminal domain results in a phenotype like that of the *AG* loss-of-function mutant (*ag*), indicating that the C-terminal domain participates in AG functions [106]. For SEP3, the interaction between helices in the N-terminal domain and those in the C-terminal domain of different partner proteins creates a hydrophobic interface that facilitates dimerization [107]. This supports the hypothesis that the C-terminal domain participates in the stabilization of protein complexes. The dimerization of TFs has an important role in regulating heterodimerization, enabling dynamic temporal responses to changes in protein concentrations, among other functions [108]. Given that most MADS-domain proteins function as homo or heterodimers [107], the role of IDRs in their C-terminal region becomes of particular interest.

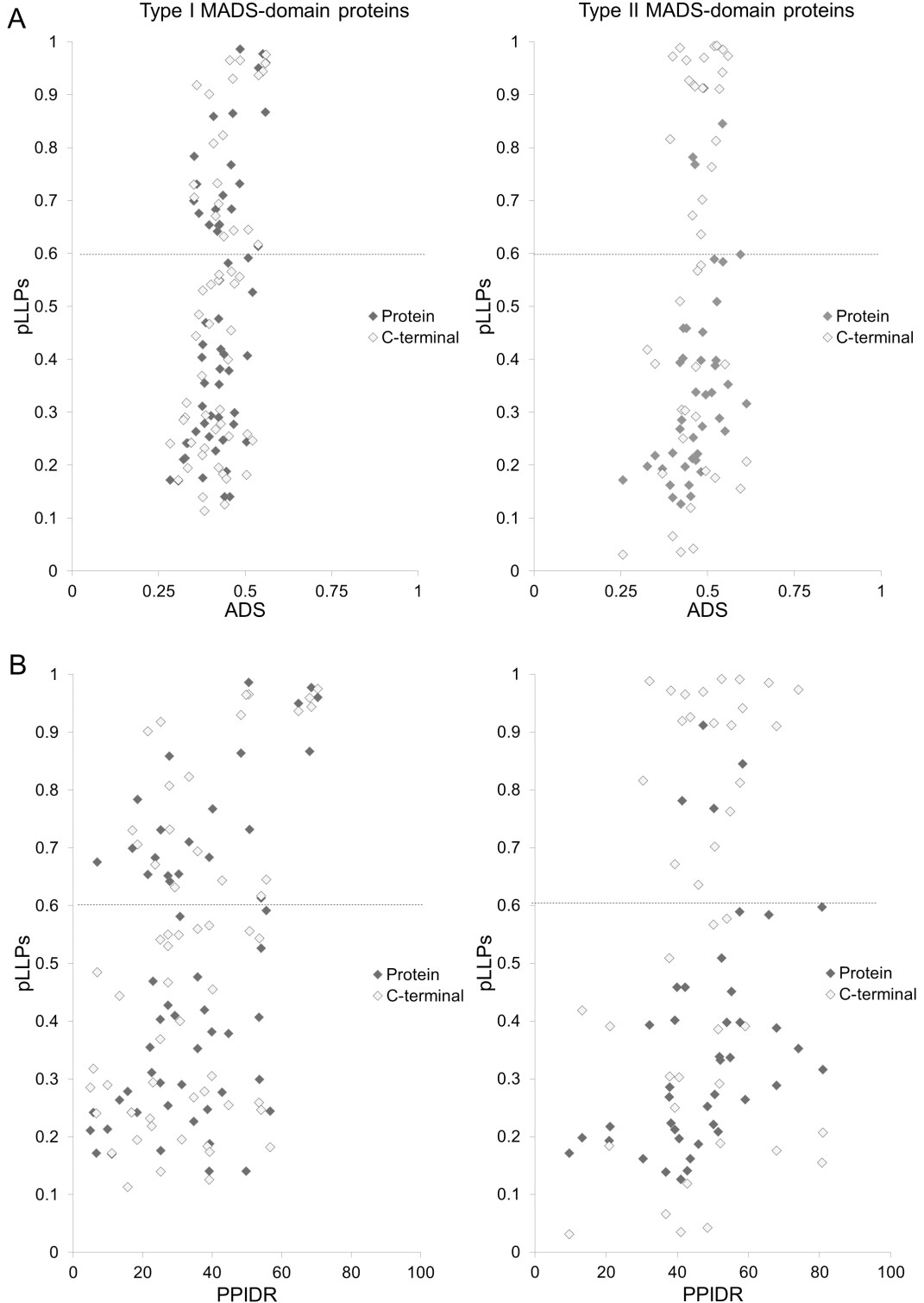

**Fig 6. Liquid-Liquid Phase Separation (LLPS) propensity in Type I (right panel) and Type II (left panel) MADS-domain proteins.** The LLPS probability index (pLLPS) was calculated using FuzPred, while the ADS and PPIDR were derived from RIDAO predictions. The scatterplots illustrate the relationship between the pLLPS index and ADS **(A)** or PPIDR **(B)** for the full-length proteins (dark diamonds) and the C-terminal (light grey diamonds), for Type I and Type II MADS-domain proteins.

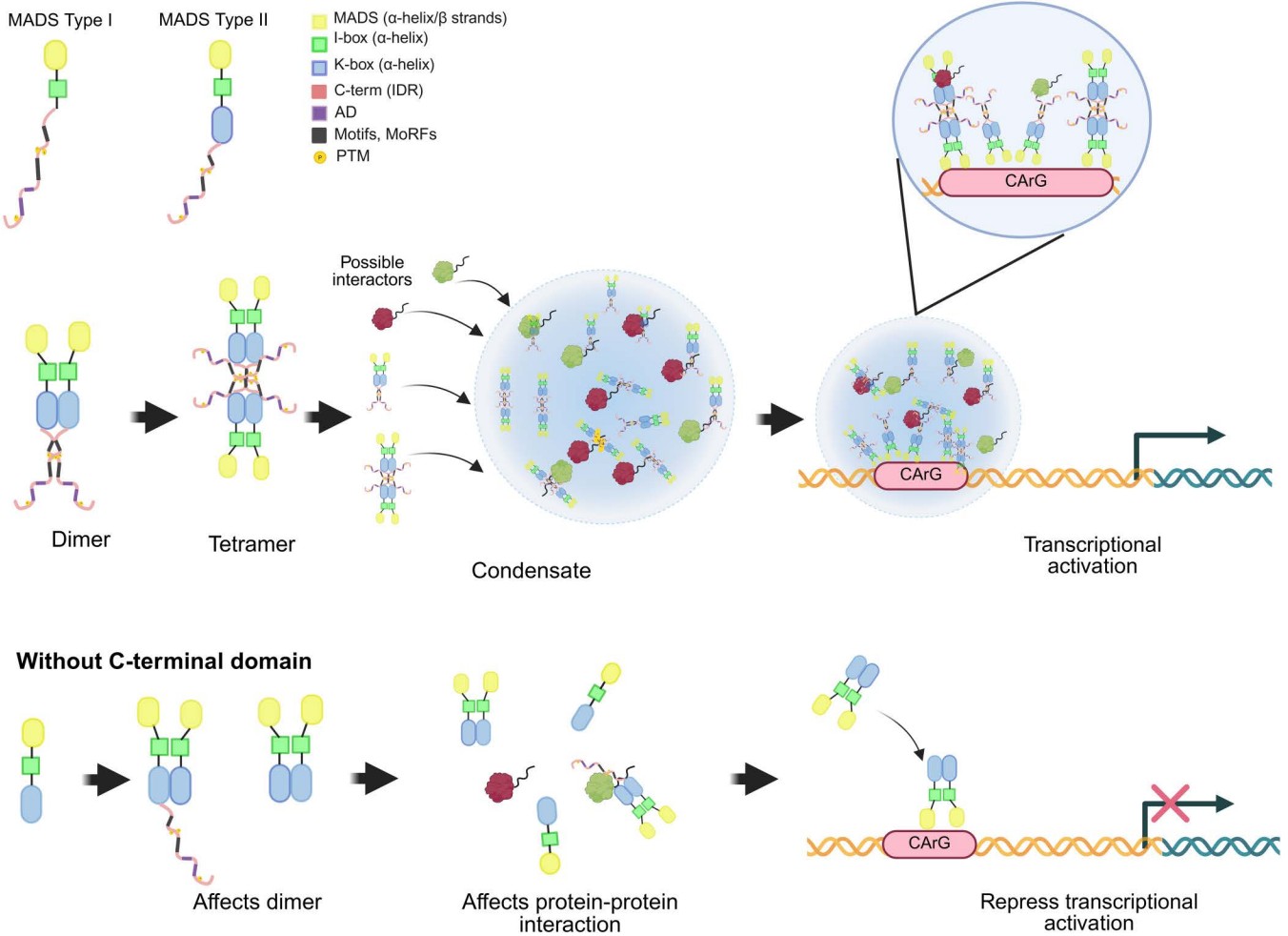

**Fig 7. Proposed model for the role of intrinsically disordered regions (IDRs) in the C-terminal region of MADS-domain transcription factors.** Top left panel – Schematic representation of Type I and Type II MADS proteins. Different functional domains are color-coded, with the C-terminal region illustrated as a red ribbon. In the C-terminal IDR are Activation Domains (ADs), Molecular Recognition Features (MoRFs) or motifs, and phosphorylation sites (see color-codes to the left of the diagrams). These features are associated with promoting protein–protein interactions both between MADS-domain proteins and with other regulatory partners. Some MADS-domain proteins display a high propensity for liquid–liquid phase separation, suggesting their involvement in the formation of biomolecular condensates. Such transcription factor condensates have been reported to enhance target gene expression and, in some cases, to recruit components of the transcriptional machinery (e.g., RNA Pol II). In some instances, deletion of the C-terminal region of MADS-domain proteins disrupts their protein–protein interactions, potentially impairing dimer formation and resulting in reduced activation of target gene activation.

The analysis conducted in this study also revealed that structural disorder in the C-terminal region of MADS-domain proteins is widely conserved across diverse taxa, including Drosophila, yeast, and human MADS-domain proteins. In humans, there are four MADS-domain proteins (MEF2A-D) mainly involved in neural development, muscle formation, heart development, and carcinogenesis. Of note, like some Arabidopsis MADS-domain proteins, the C-terminal domain of human MEF2B and MEF2D is required for transactivation [109,110]. Furthermore, phosphorylation at specific sites within the C-terminal domain of MEF2D has been shown to inhibit its transcriptional activity [111]. Consistent with these observations, our analysis also found a high frequency of predicted phosphorylation sites in the C-terminal domain of both Type I and Type II Arabidopsis MADS-domain proteins. This finding also aligns with previous observations showing a higher

phosphorylation propensity in IDRs compared to ordered regions across entire proteomes [94]. It underscores the functional relevance of these post-translational modifications and highlights the prevalence of multisite phosphorylation within disorder regions. Although some of the sites predicted from the NetPhos may not be functional and/or are still waiting for experimentally testing, the relatively balanced distribution of experimentally tested sites across both regions in Type II suggests functional relevance in both the N-terminal and C-terminal regions. Furthermore, the higher representation of sites in the N-terminal regions (M, I, and K domains) may partly reflect a historical research focus on the DNA-binding function of the M domain. On the contrary, the C-terminal region is more variable and has not been throughly studied and might play an important but underappreciated role in MADS-domain protein function. The presence of multiple phosphorylation sites in IDRs has been associated with their role in modulating gradual cellular responses and mediating protein-protein interactions, emphasizing the regulatory role of phosphorylation within disordered domains [112–115]. Further investigation into the function of the predicted phosphorylation sites within the C-terminal disordered region of MADS-domain proteins will enhance our understanding of the mechanisms by which these TFs regulate gene expression.

By searching for conserved motifs within the disordered C-terminal region of MADS-domain proteins, we identified three distinct and specific motifs in Type I and three for Type II MADS-domain proteins. Interestingly, motif 3 identified in Type I MADS-domain proteins corresponds to a conserved region previously reported by De Bodt et al. (2003) [116]. Furthermore, some of these motifs were found in all the members of different clades, suggesting that they are associated with particular functions mediated by specific interactions and shared between the related proteins. This is the case of two motifs found in Type I MADS-domain proteins that are shared among some MADS-domain proteins of Mγ and Mβ clades [24]. Motif 4 and 5, identified in Type II MADS-domain proteins (Fig 4B), is conserved across all SOC1-clade and FLC-clade proteins, respectively, highlighting its potential functional significance. Conducting motif-swapping experiments among SOC1 and FLC clade members and MADS-domain proteins that naturally lack this motif would provide valuable insights into its functional significance. The presence of clade-specific motifs in MADS-domain proteins may be attributed to the highly conserved interaction networks within different plant MADS-domain protein clades [117]. Interactions among MADS-domain proteins are restricted by their highly conserved domains in the complete proteins [117]. Conserved motifs within the variable C-terminal region identified in this study might also be involved in facilitating and stabilizing the interactions between MADS-domain proteins.

Several examples in plants demonstrate that transcriptional activation depends on the recruitment of coactivators. In MADS-domain proteins, tetramer formation increases the DNA regions available for transcriptional binding, enhancing their regulatory capacity [107,118]. Using the data from activation domains (ADs) obtained by Morffy et al. (2024) [72], we showed that a significant proportion of the potential ADs for MADS-domain TFs are in their C-terminal IDR. Moreover, we found that some of the identified ADs overlap with a conserved motif in the MADS-domain TFs of the SOC1 clade. This is particularly evident in the C-terminal of Type II MADS-domain proteins. The conserved motif in the C-terminal IDR of SEP homologs across different angiosperms [119], which corresponds to motif 4 in the SOC TFs, coincides with one of the ADs identified by Morffy et al. (2024) [72]. This overlap strongly suggests that this motif has a functional significance. We made similar observations for AP1 TFs, where ADs, characterized by the presence of acidic, proline-rich, and glutamine-rich subdomains, have been experimentally identified in their C-terminal IDRs [101].

Interestingly, Type I MADS-domain proteins exhibit more putative ADs than Type II and this could reflect specific roles for Type I MADS-domain proteins not only in the female gametophyte development but also in seed development [28,120–124]. Unfortunately, we were unable to find any study specifically analyzing the functional relevance of the C-terminal domains or any other regions of these proteins. This outcome adds further interest to the findings presented here and encourages future research into the role of the potential ADs in mediating the association between MADS-domain proteins and their co-regulators.

Finally, our analysis revealed that some MADS-domain proteins have a propensity to undergo liquid-liquid phase separation (LLPS). Nevertheless, even among those with high disorder scores, particularly within Type II MADS-domain

proteins, not all appear capable of undergoing LLPS. This discrepancy may be influenced by the fact that our analysis was based solely on protein primary structures, without accounting for possible posttranslational modifications such as phosphorylation. As we show, MADS-domain TFs could be phosphorylated at multiple sites. These posttranslational modifications, whether occurring at one or several sites, could be implicated in the promotion of LLPS, with the extent of phase separation likely dependent on specific cellular conditions. Additionally, the LLPS propensity obtained in this analysis is in agreement with findings showing that the capacity to drive LLPS is not determined solely by the disordered nature of the sequence. Instead, it depends on specific sequence features, such as the distribution and patterning of aromatic and charged residues. These sequence-encoded patterns are essential for enabling the multivalent interactions required for the formation of biomolecular condensates [125]. When only the C-terminal region of MADS-domain proteins is analyzed for LLPS propensity, the effect of intrinsic disorder shifts the scores upward. A greater number of proteins present LLPS scores above 0.6 compared to analyses of the full-length proteins, supporting the contribution of IDRs to protein condensation propensity. Furthermore, some studies have suggested that ADs can drive TF phase separation, leading to the formation of transcriptional condensates associated with chromatin [126–128]. However, recent findings indicate that this mechanism is not universal, as the recruitment of activators can also occur independently of phase separation. Important events that enhance transcriptional activation include the multivalent interactions mediated by the ADs, which increase the residence time of TFs on chromatin and thereby promote the recruitment of coactivators [97].

This study highlights common characteristics shared by MADS-domain proteins, not only in plants but also across organisms from other domains of life. Notably, the presence of a disordered C-terminal region in Type I and Type II MADS-domain TFs stands out. The functional significance of this region is strongly supported by the identification of several potential ADs and phosphorylation sites. Furthermore, we identified not only putative protein-protein interaction sites within this region but also conserved motifs specific to evolutionary related MADS-domain proteins, further supporting their role in the transcriptional regulatory function of these TFs.

The remarkable conservation of the structurally disordered C-terminal region in MADS-domain TFs suggests specific biological functions for this region. Some of these may be associated with the presence of conserved motifs and/or phosphorylation sites. However, these elements correspond to short segments within a broader region that, based on primary sequence alignments, do not appear to be under strong evolutionary constraint. To date, the persistence of IDRs across the proteomes of all analyzed organisms remains an open question. Although some evolutionary approaches have attempted to identify conserved molecular features (i.e., NCPR, kappa, FCR, etc.) in the amino acid sequences of IDRs [129], the findings suggest that, while certain molecular features are linked to known functions in yeast IDRs and may reflect a mechanism of IDR evolution, this pattern does not appear to extend to multicellular organisms like *Drosophila* [130]. These observations show that although IDRs may follow unique patterns of amino acid substitutions, intrinsic disorder itself is subjected to dynamic evolutionary processes, shaped by more complex evolutionary constraints across evolving properties of different domains of life.

Considering the well-established role of MADS-domain TFs in regulating diverse developmental processes and stress responses, the conservation of a structurally disordered C-terminal region across all family members, along with the presence conserved motifs, potential activation domains and phosphorylation sites, suggest a shared regulatory mechanism. Based on these findings, we proposed a mechanistic model describing the functional role of specific structural elements within the C-terminal domain of both Type I and Type II MADS-domain TFs. The structural flexibility provided by the IDRs in the C-terminal domain, combined with the presence of MoRFs, ADs, and a high phosphorylation propensity, points to a regulatory role in modulating the MADS-domain TF activity within transcriptional complexes (Fig 7, S6 Table). In this model, these IDRs confer both structural flexibility and modularity, facilitating the formation of dynamic protein complexes. This may occur either through a high propensity for liquid-liquid phase separation (LLPS) or by promoting transient interactions with other proteins, both of which support the formation of transcriptional condensates (Fig 7). In the present study, we identified MoRFs and ADs within the C-terminal IDRs, where MoRFs likely mediate specific,

transient interactions with regulatory proteins or other MADS-domain TFs in concert with ADs, these elements may modulate the assembly and stability of transcriptional complexes. Previous reports suggest that the formation of condensates enriched in transcription factors enables fine-tuned regulation of transcriptional activity, particularly in response to physiological or developmental cues [131,132]. Within such condensates, a variety of proteins can interact with TF IDRs through their ADs and LLPS-driven mechanisms further supporting the functional significance of the C-terminal IDRs in MADS-domain TFs.

Overall, this study provides valuable insights for a deeper understanding of the relationship between protein structure and function for MADS-domain TFs. We believe this information will encourage and support further experimental studies by researchers working on MADS-domain proteins in diverse biological systems, especially to investigate the functional relevance of these conserved regions.

## Supporting information

**S1 Table. MADS-domain proteins of Arabidopsis, other plant species, and non-plant species used in this study.** (DOCX)

**S2 Table. MADS-domain proteins analyzed by Morffy et al. (2024) [64].** AD = activation domain. AD, Maybe and Not AD definitions are given based on the PADI (plant activation domain identification). PADI score ≥1 and mean disorder >0.5 are defined as "AD"; PADI score ≥1 and mean disorder de ≤ 0.5 are defined as "Maybe"; PADI <1 are defined as "Not AD". (DOCX)

**S3 Table. R packages used in this research.** These packages are available from CRAN (https://CRAN.R-project.org/) or Bioconductor (Huber et al., 2015). (DOCX)

**S4 Table. Arabidopsis MADS-domain proteins, disordered regions and phosphorylation sites.** (XLSX)

**S5 Table. Experimentally tested phosphorylation sites obtained from two different databases Athena (https://athena.proteomics.wzw.tum.de/master_arabidopsisshiny/), and EPSD (https://epsd.biocuckoo.cn/Browse.php).** Sites were mapped on domains, according to UniProt and Liu et al., 2021 definition. See main text for a detailed description. (DOCX)

**S6 Table. Experimental evidence supporting the functional relevance of the C-terminal region in MADS-domain proteins across different species.** (DOCX)

**S1 Fig. Physicochemical properties of Type I and Type II MADS-domain proteins. (A)** Net charge per residue (NCPR), **(B)** Charge distribution (Kappa), and **(C)** Fraction of Charged Residues (FCR). Whiskers indicate ±1.5*IQR based on Tukey test. The middle line represents the median. (TIF)

**S2 Fig. A complete Maximum Likelihood (ML) phylogeny of Arabidopsis MADS-domain proteins including MADS-domain proteins of other plant species and non-plant species.** *Oryza sativa japonica* (*Os*), *Solanum dulcamara* (*SD*), *Mangifera indica* (*MI*), *Populus trichocarpa* (*PT*), *Amborella trichopoda* (*AMB*), *Pinus radiata* (*Prad*), *Selaginella mollendorffii* (*SM*), *Chara braunii* (*CB*), *Chlorella dessiccata* (*CD*), *Saccharomyces cerevisiae* (*SC*), *Drosophila melanogaster* (*DM*), *and Homo sapiens* (*HS*). Yellow-shaded branches cover Type I MADS grouped with SRF-like and MEF-like MADS-domain proteins. Purple-shaded branches cover most Type II MADS-domain proteins. Numbers adjacent

to nodes represent bootstrap support. Orange dots at particular nodes indicate the putative ancestral motif for that specific clade.
(TIF)

**S3 Fig. Structural disorder in the C-terminal region and the full-length proteins of MADS-domain transcription factors from different organisms.** Boxplots showing the ADS and PPIDR values for the C-terminal region and the full-length proteins from: **(A)** Plant species analysed in this study, excluding Arabidopsis. **(B)** *Saccharomyces cerevisiae*, **(C)** *Homo sapiens*, and **(D)** *Drosophila melanogaster*. Whiskers indicate ±1.5*IQR according to the Tukey test. The middle line represents the median.
(TIF)

**S4 Fig. Distinctive motifs in MADS-domain proteins of SOC1 (A) and FLC (B) clades.** The phylogenetic tree was derived from the complete MADS-domain protein tree shown in Supplementary Fig 2. To enhance the visualization of phylogenetic relationships among the proteins, branch lengths were rescaled and truncated. The conserved amino acid residues of the SOC1-motif and FLC-motif are highlighted in bold within the consensus motif.
(TIF)

**S5 Fig. Identified MoRFs in MADS-domain proteins.** MADS-domain protein sequences from Arabidopsis, other plants, and non-plant organisms are shown, with predicted MoRFs highlighted in yellow Putative MoRFs were identified using the fmoRFpred algorithm [123], based on the analysis of full-length protein sequences.
(TIF)

## Acknowledgments

The first authors wish to thank Consejo Nacional de Humanidades, Ciencias y Tecnología (CONAHCyT) for the postdoctoral scholarships granted (E.R.A., CVU number 413896; T.N.R., CVU 501149).

## Author contributions

**Conceptualization:** Adriana Garay-Arroyo, Alejandra A. Covarrubias.

**Data curation:** Erandi Ramírez-Aguirre.

**Formal analysis:** Alejandra A. Covarrubias, Erandi Ramírez-Aguirre, Teresa Beatriz Nava-Ramírez.

**Funding acquisition:** Adriana Garay-Arroyo.

**Investigation:** Adriana Garay-Arroyo, Alejandra A. Covarrubias, Erandi Ramírez-Aguirre, Teresa Beatriz Nava-Ramírez.

**Methodology:** Erandi Ramírez-Aguirre, Teresa Beatriz Nava-Ramírez.

**Supervision:** Adriana Garay-Arroyo, Alejandra A. Covarrubias.

**Visualization:** Erandi Ramírez-Aguirre, Teresa Beatriz Nava-Ramírez.

**Writing – original draft:** Adriana Garay-Arroyo, Alejandra A. Covarrubias, Erandi Ramírez-Aguirre, Teresa Beatriz Nava-Ramírez.

**Writing – review & editing:** Adriana Garay-Arroyo, Alejandra A. Covarrubias, Erandi Ramírez-Aguirre, Teresa Beatriz Nava-Ramírez.

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
