## [Decision Letter · Decision Letter 0]

30 Apr 2025

Dear Dr. Garay Arroyo,

Thank you for submitting your manuscript to PLOS ONE. After careful consideration, we feel that it has merit but does not fully meet PLOS ONE’s publication criteria as it currently stands. Therefore, we invite you to submit a revised version of the manuscript that addresses the points raised during the review process.

We look forward to receiving your revised manuscript.

Kind regards,

Ajit Prakash, PhD

Academic Editor

PLOS ONE

Journal Requirements:

2. Thank you for stating in your Funding Statement: [This work was partially financed by the projects PAPIIT No. IN213524 from Universidad Nacional Autónoma de México (UNAM), and project No. CBF2023-2024-1002 from the Consejo Nacional de Humanidades, Ciencias y Tecnologías

(CONAHCyT) granted to A.G.A.; and project No. CF2023-I-503 from the Consejo

Nacional de Humanidades, Ciencias y Tecnologías (CONAHCyT) granted to A.A.C.].

3. Please update your submission to use the PLOS LaTeX template. The template and more information on our requirements for LaTeX submissions can be found at http://journals.plos.org/plosone/s/latex

4. Please upload a new copy of Figure S5 as the detail is not clear. Please follow the link for more information: https://blogs.plos.org/plos/2019/06/looking-good-tips-for-creating-your-plos-figures-graphics/"" https://blogs.plos.org/plos/2019/06/looking-good-tips-for-creating-your-plos-figures-graphics/

5. Thank you for stating the following financial disclosure: [This work was partially financed by the projects PAPIIT No. IN213524 from Universidad Nacional Autónoma de México (UNAM), and project No. CBF2023-2024-1002 from the Consejo Nacional de Humanidades, Ciencias y Tecnologías

(CONAHCyT) granted to A.G.A.; and project No. CF2023-I-503 from the Consejo

Nacional de Humanidades, Ciencias y Tecnologías (CONAHCyT) granted to A.A.C.].

Additional Editor Comments:

Please address the concerns raised by reviewers and provide point to point response.

Reviewers' comments:

Reviewer's Responses to Questions

**Comments to the Author**

1. Is the manuscript technically sound, and do the data support the conclusions?

Reviewer #1: Yes

Reviewer #2: Yes

2. Has the statistical analysis been performed appropriately and rigorously?

Reviewer #1: Yes

Reviewer #2: No

3. Have the authors made all data underlying the findings in their manuscript fully available?

Reviewer #1: Yes

Reviewer #2: Yes

4. Is the manuscript presented in an intelligible fashion and written in standard English?

Reviewer #1: No

Reviewer #2: Yes

Reviewer #1: The authors have explored the role of intrinsically disordered regions (IDRs) in transcription factors, focusing on MADS-domain proteins in Arabidopsis thaliana. IDRs contribute to protein-protein interactions and functional flexibility. In this study via bioinformatic analysis revealed that most Arabidopsis MADS-domain proteins have IDRs in their C-terminal region, with Type II proteins exhibiting more disorder than Type I. Similar IDRs were found in orthologous proteins from non-plant species (Drosophila, Saccharomyces cerevisiae, and Homo sapiens), often with longer disordered regions. Conserved motifs and putative activation domains were identified in Arabidopsis MADS-domain proteins, suggesting interactions with co-regulatory partners. These findings highlight the evolutionary conservation of structural disorder in MADS-domain proteins and its role in transcriptional regulation.

Comments:

I think one of the main concerns of the current submission is more editorial. The manuscript was heavily disorganized, especially with figure legends. It was oddly placed and poorly labeled.

Figure 1) In the box plot showing MDP for Type 1 MAD domain, there is no significant difference between complete protein and C-terminal region like shown in Type 2. The authors have not commented on how that impacts the conclusion on C-terminal IDPs. Does that effect the difference in the protein length and RIDAO Map score.

Figure 2) The pattern of phosphorylation sites in Type 1 and Type 2 different. Phosphorylation can be a key event in stabilizing proteins, and based on the data on Type 2 have higher propensity of phosphorylation in c-terminal and with a small incremental effect. This does not alone support the conclusion that c -terminal of MADS protein have higher phosphorylation site, there should be more analysis done. Comparing non-MADS IDPs can be useful.

Figure 5) PADI score is dependent on sequence composition, supporting this analysis with different PADI threshold and performance matrix with sensitivity, specificity, ppv, npv with same data can increase the confidence on this analysis.

Figure 6) IDPs drive condensate formation and Type 2 have more MDP index and correlation in figure 6 how Type 1 have more LLPS forming propensity than Type 2. The authors have addressed the discrepancy in discussion. It would add more value if there is more similar analysis on known IDPs and LLP forming index. Additionally what if they only input C-terminal region , will that change the LLPS forming Index ?

The discussion is really long, it will help the manuscript if the authors try to keep it more concise in general.

Reviewer #2: This manuscript presents a bioinformatic analysis of intrinsically disordered regions in MADS-domain transcription factors across diverse taxa. The authors demonstrate that C-terminal disorder is consistently present in both plant and non-plant MADS-domain proteins, with Type II proteins showing higher overall disorder than Type I proteins. They identify conserved motifs within these disordered regions that appear clade-specific, along with overlapping Molecular Recognition Features particularly in Type II proteins. The study further characterizes potential activation domains and liquid-liquid phase separation propensities, finding more activation domains in Type I proteins. While the conservation of these structural features across evolutionary distant taxa strongly suggests functional importance in transcriptional regulation, I have some concerns about the exclusively computational nature of the analysis. The study would benefit from experimental validation of key predictions, clearer distinction between correlation and causality in evolutionary conservation, more robust statistical analysis, and a more integrated model explaining how these various structural features collectively contribute to MADS-domain protein function.

Concern(s)/ Limitations and comments:

- My core concern is about the experimental validation. The study relies exclusively on bioinformatic predictions without any experimental validation of the identified IDRs, MoRFs, or activation domains. While computational approaches provide valuable insights, experimental confirmation is needed to significantly strengthen the findings. I would suggest authors include at least some experimental validation for key findings, particularly for the identified motifs and their potential functional roles. Techniques such as circular dichroism (CD) spectroscopy could confirm the disordered nature of the C-terminal regions, while yeast two-hybrid or co-immunoprecipitation assays could validate protein-protein interactions mediated by the identified MoRFs. Moreover, directed mutagenesis of conserved motifs followed by functional assays would help establish their biological significance, as demonstrated by Cho et al. (1999) for the C-terminal region of APETALA1.

- While the conservation of structural disorder across taxa is interesting, the manuscript doesn't establish whether this conservation reflects functional constraints or simply represents a structural consequence of sequence evolution. Further discussion on this point would strengthen the paper. As noted by Pancsa and Tompa (2012), intrinsic disorder can be maintained through both purifying selection and neutral evolution. The authors could address this by analyzing the rates of synonymous versus non-synonymous substitutions in the disordered regions compared to ordered domains, which would provide evidence for selective pressure maintaining these IDRs. Additionally, examining the conservation of physicochemical properties rather than just sequence could help distinguish between functional conservation and structural coincidence.

- Although the manuscript speculates on the functional significance of the identified motifs and structural features, it doesn't provide concrete evidence for their roles. I would suggest exploring how these findings connect to known functional differences between Type I and Type II MADS-domain proteins. For instance, the predominance of activation domains in Type I proteins could be related to their specific roles in gametogenesis, while the different patterns of MoRFs in Type II proteins might explain their more diverse developmental functions. Supporting this connection with existing phenotypic data from mutant studies would significantly strengthen the biological relevance of the identified structural features. The recent work by Morffy et al. (2024) on plant transcriptional activation domains could provide a framework for such functional integration.

- While the paper identifies various structural features (IDRs, MoRFs, ADs), it doesn't fully integrate these findings into a cohesive model of how these features might work together to influence protein function. I believe the authors should develop a unified mechanistic model showing how the identified structural elements (IDRs, MoRFs, and ADs) work together to regulate MADS-domain protein function in transcriptional complexes, similar to the framework proposed by Wright and Dyson (2015) for IDR-mediated assembly of transcriptional machinery.

- I believe the comparison between plant and non-plant MADS-domain proteins needs to be expanded to better understand the evolutionary conservation and divergence of these structural features. The current analysis shows that non-plant MADS-domain proteins have longer IDRs, but doesn't explore whether this reflects different functional constraints or adaptations. I suggest including a more detailed phylogenetic analysis that traces the evolution of specific structural features across lineages, similar to the approach taken by Davey et al. (2012) for short linear motifs. This could reveal whether certain structural properties emerged independently in different lineages or were present in the common ancestor of all MADS-domain proteins.

- I believe the statistical analyses need strengthening, especially when comparing Type I and Type II proteins. While Wilcoxon tests are used appropriately, the authors should implement corrections for multiple testing and address how the unequal sample sizes (58 Type I vs. 42 Type II proteins) might affect statistical power. I recommend incorporating bootstrapping or permutation tests to increase confidence in the reported differences, following approaches described by Efron and Tibshirani (1994) for biological data analysis.

**Do you want your identity to be public for this peer review?** For information about this choice, including consent withdrawal, please see our Privacy Policy

Reviewer #1: No

Reviewer #2: **Yes: ** Mohammad Ashhar Iqbal Khan

---

## [Author Response · Author response to Decision Letter 1]

4 Jul 2025

We thank the reviewers for their valuable comments and feedback, which have undoubtedly helped us improve the manuscript. Thanks to your comments and in the interest of greater clarity, we have decided to include both disorder scores retrieved from the RIDAO platform in the intrinsic disorder analysis: the Average Disorder Score (ADS) and the Percentage of Predicted Disordered Residues (PPDR). In the originally submitted version of the manuscript, we only included the PPDR (MDP), which corresponds to PPDR or Per-MDP in the RIDAO platform. According to RIDAO, PPDR represents the proportion of amino acids with a disorder above 0.5, indicating significant intrinsic disorder, whereas ADS reflects the overall disorder propensity of the protein. As expected, both parameters show a positive correlation. The inclusion of both metrics reinforces the evidence for the intrinsic disorder propensity obtained in the analyzed MADS domain TFs.

Reviewer #1: The authors have explored the role of intrinsically disordered regions (IDRs) in transcription factors, focusing on MADS-domain proteins in Arabidopsis thaliana. IDRs contribute to protein-protein interactions and functional flexibility. In this study via bioinformatic analysis revealed that most Arabidopsis MADS-domain proteins have IDRs in their C-terminal region, with Type II proteins exhibiting more disorder than Type I. Similar IDRs were found in orthologous proteins from non-plant species (Drosophila, Saccharomyces cerevisiae, and Homo sapiens), often with longer disordered regions. Conserved motifs and putative activation domains were identified in Arabidopsis MADS-domain proteins, suggesting interactions with co-regulatory partners. These findings highlight the evolutionary conservation of structural disorder in MADS-domain proteins and its role in transcriptional regulation.

Comments:

- I think one of the main concerns of the current submission is more editorial. The manuscript was heavily disorganized, especially with figure legends. It was oddly placed and poorly labeled.

R: Thank you for this observation. The Figure legends were included in the manuscript according to the journal´s instructions. However, as requested by the reviewer, we have included them after the references and have improved their writing to enhance readability. Additionally, we have thoroughly revised the manuscript to improve organization.

Figure 1) In the box plot showing MDP for Type 1 MAD domain, there is no significant difference between complete protein and C-terminal region like shown in Type 2. The authors have not commented on how that impacts the conclusion on C-terminal IDPs. Does that effect the difference in the protein length and RIDAO Map score.

R: The lack of differences in the disorder index between the C-terminal domain and the complete Type I MADS-domain proteins reflects the relative proportion of the C-terminal domain within the complete protein. Type I MADS-domain proteins are generally longer than Type II, and their C-terminal regions represent a significantly larger proportion of their total sequence. Therefore, the disorder scores of the C-terminal domain contribute substantially to the overall disorder scores of the entire protein.

-Figure 2) The pattern of phosphorylation sites in Type 1 and Type 2 different. Phosphorylation can be a key event in stabilizing proteins, and based on the data on Type 2 have higher propensity of phosphorylation in c-terminal and with a small incremental effect. This does not alone support the conclusion that c -terminal of MADS protein have higher phosphorylation site, there should be more analysis done. Comparing non-MADS IDPs can be useful.

R: We agree with the reviewer’s comment. Analysis of the phosphorylation propensity in a protein sequence does not assure that the predicted sites will be phosphorylated. However, there are several reports supporting that intrinsically disordered regions contain a larger proportion of potential phosphorylation sites than those regions with an ordered and stable structure (Iakoucheva et al. 2004 DOI: 10.1093/nar/gkh253; Gao and Xu, 2012 PMID: 22174266; Marijn and Ott, 2012 doi.org/10.3389/fpls.2012.00086; Koike et al., 2019 doi.org/10.1002/pro.3789; Usher et al., 2024 https://doi.org/10.1016/j.bpj.2024.10.021; Holehouse and Kragelund, 2024 doi.org/10.1038/s41580-023-00673-0). This finding makes sense given the higher exposition of potential phosphorylation sites in more flexible regions. Additionally, it has been found that the amino acid composition, sequence complexity, hydrophobicity, charge, and other sequence features surrounding confirmed phosphorylation sites closely resemble those characteristics of IDRs (Iakoucheva et al. 2004, DOI: 10.1093/nar/gkh253 and by Koike et al., 2019, DOI:10.1002/pro.3789).

The analysis suggested by the reviewer with a more extensive set of proteins has been done by Iakoucheva et al. (2004; DOI: 10.1093/nar/gkh253) using different proteomes. These authors have used complete proteomes, finding that proteins containing IDRs show a higher phosphorylation propensity than ordered proteins. We integrated this information into the manuscript (see Discusion, page 19, lines 523 – 542)

Because most of the proposed sites have not been experimentally validated, we have not stated that this is the case. Most of the validated phospho-sites are in the MADS-domain proteins' conserved motifs, and these experiments have been conducted under specific developmental stages and/or growth conditions, leaving aside alternative or additional phosphorylations under different experimental conditions. We think that the trend of these analyses is because, in general, the MADS’ IDRs have not received enough attention.

This is more evident when we look at the number of validated phosphorylation sites in MADS-domain Type II TFs, the best-characterized of this family. Type II MADS TFs show a similar number of phospho-sites in both the N- and C-terminal regions. This finding contrasts with the data on MADS-domain Type I TFs, where the validated phospho-sites are more abundant in the N-terminal region (see new Supplementary Table S5). Therefore, we think that the data in our manuscript provides an invitation to perform more methodical studies on MADS IDRs to find out the functional relevance of these regions, which are highly represented in this protein family.

Additionally, we have added a new paragraph in the phosphorylation results (Page 14, lines 343-351) that compares the experimentally validated phosphorylation patterns between Type I and Type II MADS-domain proteins (see new Supplementary Table S5).

We have carefully revised the manuscript, ensuring that the analysis we present in this manuscript refers to propensities, based on validated bioinformatic tools.

Figure 5) PADI score is dependent on sequence composition, supporting this analysis with different PADI threshold and performance matrix with sensitivity, specificity, ppv, npv with same data can increase the confidence on this analysis.

R: As mentioned in the manuscript, the data on plant activation domain identification (PADI) were directly obtained from Morffy et al. (2024; DOI: 10.1038/s41586-024-07707-3). In that study, the authors experimentally identified activation domains in various plant TFs using a comprehensive library. This library consisted of overlapping 40 amino acid fragments, spanning the entire set of plant TFs, with a step size of 10 amino acids, resulting in a total of 68,441 fragments. These fragments were screened in yeast to assess their transcriptional activation capacity. Based on this experiment and subsequent normalization, a PADI score was assigned to each fragment. Among the fragments showing transcriptional activation activity, some corresponded to MADS-domain TFs. The PADI scores included in our manuscript are those reported in this study. The localization of activation domains (AD) within MADS-domain TFs was inferred from the results obtained in Morffy et al. (2024), where the authors applied a neural network-based algorithm, known as transcriptional activation domain activity (TADA) network. Their work integrated multiple layers of analysis, including the construction of a feature matrix and the use of methods to assess the impact of both individual input features and border local and global interactions predicted by TADA. Additionally, these results were further analyzed using deep learning to identify key properties relevant to the prediction of ADs. A SHAP analysis was applied to capture non-linear and linear correlations, thereby uncovering complex patterns. This was followed by additional deep-learning steps that culminated in the development of a tool capable of predicting potential ADs. We have included part of this information in the Materials and Methods section. Page 8-9 lines 195-218.

As the reviewer may notice, modifying each of the many steps involved in calculating the propensity of specific sequences to function as ADs would be a highly complex task. We consider that such efforts are beyond the scope of the present manuscript.

- Figure 6) IDPs drive condensate formation and Type 2 have more MDP index and correlation in figure 6 how Type 1 have more LLPS forming propensity than Type 2. The authors have addressed the discrepancy in discussion. It would add more value if there is more similar analysis on known IDPs and LLP forming index.

R: Three different computational approaches were employed to evaluate the LLPS propensity index: FuzPred, PSPredict, and PSHunter. The table containing the values obtained for the full-length proteins and their corresponding C-terminal regions is included as part of the response to the reviewers (pLLPs software comparison in a reviewer's specific file). Values indicating a high propensity for liquid–liquid phase separation, according to the criteria specified by each prediction tool, are highlighted in yellow.

Among these, the analysis performed with FuzPred revealed a higher LLPS propensity in comparison to the other two predictors, although all three identified certain proteins with the potential to form biomolecular condensates.

The three tools are designed to predict the likelihood of a protein sequence undergoing liquid-liquid phase separation, particularly in the context of intrinsically disordered regions, but PSPredict and PSHunter exhibit some limitations. Specifically, these methods do not explicitly account for conformational dynamics or contextual interaction features, and they have reduced resolution in capturing the local contributions of specific sequence segments (Chu et al., 2022, doi.org/10.1186/s12859-022-04599-w; Sun et al., 2024, /doi.org/10.1038/s41467-024-46901-9).

In contrast, FuzPred incorporates the prediction of conformational fuzziness, referring to the propensity of a disordered region to retain dynamic and structurally adaptable behavior even upon molecular interaction. This approach explicitly considers both multivalency and structural flexibility as key drivers of LLPs, thereby allowing for the identification of functionally relevant interaction-prone regions that may not be detected by more generalized predictors. For this reason, FuzPred was selected as the primary tool for assessing LLPS propensity in our study (Hatos et al., 2023; doi: 10.1093/nar/gkad214).

Additionally, what if they only input C-terminal region, will that change the LLPS forming Index?

R: To address the reviewer’s question, which may also be shared by readers, we have included in the revised Figure 6 the data corresponding to the C-terminal regions of the MADS domain TFs. As shown in the updated figure, and as expected, the LLPS propensity indices for these regions are generally higher than those observed for the full-length proteins. This supports the notion that the C-terminal regions contribute significantly to LLPS propensity, and that the rest of the protein sequences may also influence these predictions. Consequently, the LLPS propensity of the full-length proteins does not always correlate with that of their C-terminal regions.

As noted in our answer to the previous comment, most LLPS predictors focus on the properties of IDRs, as these regions are key facilitators of protein-protein interactions, some of which drive LLPS through homotypic interactions among disordered segments. Moreover, IDRs can mediate interactions with other proteins, further promoting condensate formation (Xu et al., 2024; doi: 10.1073/pnas.2407633121).

However, the capacity to drive LLPS is not determined solely by the disordered nature of the sequence. Instead, it depends on specific sequence features, such as the distribution and patterning of aromatic and charged residues. These sequence-encoded patterns are essential for enabling the multivalent interactions required for the formation of biomolecular condensates (Borcherds et al., 2021; doi: 10.1016/j.sbi.2020.09.004).

We have added a comment on this point in the Discussion section of the manuscript (Page 21-22 lines 586-612).

The discussion is really long; it will help the manuscript if the authors try to keep it more concise in general.

R: As requested by the reviewer, we have made an effort to reduce the length of the Discussion section. However, this has proven challenging due to the simultaneous request to include additional information.

Reviewer #2: This manuscript presents a bioinformatic analysis of intrinsically disordered regions in MADS-domain transcription factors across diverse taxa. The authors demonstrate that C-terminal disorder is consistently present in both plant and non-plant MADS-domain proteins, with Type II proteins showing higher overall disorder than Type I proteins. They identify conserved motifs within these disordered regions that appear clade-specific, along with overlapping Molecular Recognition Features, particularly in Type II proteins. The study further characterizes potential activation domains and liquid-liquid phase separation propensities, finding more activation domains in Type I proteins. While the conservation of these structural features across evolutionary distant taxa strongly suggests functional importance in transcriptional regulation, I have some concerns about the exclusively computational nature of the analysis. The study would benefit from experimental validation of key predictions, a clearer distinction between correlation and causality in evolutionary conservation, more robust statistical analysis, and a more integrated model explaining how these various structural features collectively contribute to MADS-domain protein function.

Concern(s)/ Limitations and comments:

- My core concern is about the experimental validation. The study relies exclusively on bioinformatic predictions without any experimental validation of the identified IDRs, MoRFs, or activation domains. While computational approaches provide valuable insights, experimental confirmation is needed to significantly strengthen the findings. I would suggest authors include at least some experimental validation for key findings, particularly for the identified motifs and their potential functional roles. Techniques such as circular dichroism (CD) spectroscopy could confirm the disordered nature of the C-terminal regions, while yeast two-hybrid or co-immunoprecipitation assays could validate protein-protein interactions mediated by the identified MoRFs. Moreover, directed mutagenesis of conserved motifs followed by functional assays would help establish their biological significance, as demonstrated by Cho et al. (1999) for the C-terminal region of APETALA1.

R: We agree with the reviewer’s comment that experimental confirmation would add valuable support to our findings. However, such validation would require a substantial investment of time and resources, which will significantly delay the dissemination of the information presented in this manuscript to the scientific community interested in this important group of TFs. Moreover, this is particularly important because the present study provides, for the first time, evidence that the presence of IDRs in the C-terminal region of MADS-domain TFs is a conserved feature. Furthermore, our results suggest that this structural feature is not exclusive to

---

## [Decision Letter · Decision Letter 1]

27 Jul 2025

Structural disorder and distinctive motifs in the C-terminal region of the MADS-domain transcription factors are conserved across diverse taxa

PONE-D-25-10320R1

Dear Dr. Garay Arroyo,

We’re pleased to inform you that your manuscript has been judged scientifically suitable for publication and will be formally accepted for publication once it meets all outstanding technical requirements.

Kind regards,

Ajit Prakash, PhD

Academic Editor

PLOS ONE

Additional Editor Comments (optional):

Reviewers' comments:

Reviewer's Responses to Questions

**Comments to the Author**

Reviewer #1: All comments have been addressed

Reviewer #2: All comments have been addressed

2. Is the manuscript technically sound, and do the data support the conclusions?

Reviewer #1: Yes

Reviewer #2: Yes

3. Has the statistical analysis been performed appropriately and rigorously?

Reviewer #1: I Don't Know

Reviewer #2: Yes

4. Have the authors made all data underlying the findings in their manuscript fully available?

Reviewer #1: Yes

Reviewer #2: Yes

5. Is the manuscript presented in an intelligible fashion and written in standard English?

Reviewer #1: Yes

Reviewer #2: Yes

Reviewer #1: The authors have addressed all the comments and the authors have also provided evidence from other studies to support the questions they could have not addressed.

Reviewer #2: (No Response)

**Do you want your identity to be public for this peer review?** For information about this choice, including consent withdrawal, please see our Privacy Policy

Reviewer #1: No

Reviewer #2: **Yes: ** Mohammad Ashhar Iqbal Khan

---

## [Editor Report · Acceptance letter]

PONE-D-25-10320R1

PLOS ONE

Dear Dr. Garay-Arroyo,

I'm pleased to inform you that your manuscript has been deemed suitable for publication in PLOS ONE. Congratulations! Your manuscript is now being handed over to our production team.

Kind regards,

on behalf of

Dr. Ajit Prakash

Academic Editor

PLOS ONE